# Effects of preharvest sprays of iodine, selenium and calcium on apple biofortification and their quality and storability

**Paweł Wójcik** *

Department of Fruit Crop Management and Plant Nutrition, The National Institute of Horticultural Research, Skierniewice, Poland

* pawel.wojcik@inhort.pl

**Data Availability Statement:** All relevant data are within the paper.

**Funding:** PW; grant funded by the Polish Ministry of Science and Higher Education, no 4.1.2. The funders had no role in study design, data collection

## Abstract

The low dietary intake of iodine (I) and selenium (Se) by humans leads to serious health and socioeconomic problems. Therefore, enrichment of plants with I and Se using fertilisers containing these micronutrients is commonly recommended. In this study, we examined the impacts of combined spraying of I as iodide or iodate, Se as selenite or selenate, and calcium (Ca) as Ca-chloride on the enrichment of 'Red Jonaprince' (*Malus domestica* Borth.) apples, as well as fruit quality and their storability. Sprays were applied 2 weeks before harvest at rates of 0.5 kg I, 0.25 kg Se and 7 kg Ca per ha. Trees not sprayed with these nutrients served as controls. The tested sprays caused leaf burn, but they did not affect the cold injury of buds and shoots. Those sprays had no effect on yield, fruit size and russeting or skin colouring. At harvest, sprayed apples contained about 50 times more I and Se and 30% more Ca than the control fruit. After storage, compared to the control fruit, sprayed apples were firmer, had more organic acids and were less susceptible to disorders, such as bitter pit, internal breakdown and decay caused by *Neofabraea* spp. The results indicate that preharvest spraying with I, Se and Ca at high rates can be recommended to effectively enrich apples with I and Se and to simultaneously improve their storability.

## 1. Introduction

Humans require at least 22 minerals to meet their metabolic needs [1]. Most of these nutrients must be delivered regularly with food at all stages of human life [2]. However, the quantities in consumed food and/or their nutritive values do not always guarantee adequate amounts of available nutrients, especially when animal-derived products are excluded/limited from the diet [3].

Iodine (I) and selenium (Se) are essential trace nutrients for humans. The role of I is related mainly to the synthesis of thyroid hormones and its deficiency leads to the appearance of endemic goitres, as well as the impairment of growth, reproductive and mental functions [4]. Selenium possesses immunomodulatory effects and its deficiency causes cardiovascular diseases and many types of cancers [5]. Currently, there is an ongoing debate about the effect of

and analysis, decision to publish, or preparation of the manuscript.

**Competing interests:** The author has declared that no competing interests exist.

Se status on human response to severe acute respiratory syndrome coronavirus 2 (SARS-CoV-2) infection [6]. Given that 15% and 30% of the world's people suffered from deficiencies of Se and I, respectively [7], malnutrition problems with these micronutrients determine serious global economic and social consequences.

One of the primary methods limiting human malnutrition with I and Se is agronomic plant biofortification, mainly through soil and/or foliar fertilisation with these nutrients [8]. This biofortification method has been tested for many plant species grown under various conditions [7,9–12]. Fertilisation with I and Se have been successful in enriching edible parts of plants with these nutrients, although their efficiencies depend on the species/variety, cultivation technology, growth conditions related to climate and growing medium, mode and application technique of fertilisers, and nutrient rate used and its chemical form. According to White and Broadley [7] and Cakmak et al. [13], foliar fertilisation with micronutrients (including I and Se) should be preferred, at least for field crops with edible parts above the soil surface because the utilisation efficiencies of leaf-applied nutrients by plants are greater than those for soil-used nutrients.

Among deciduous fruit trees grown in temperate climate, the effects of agronomic biofortification with I and/or Se have been examined for plum (*Prunus domestica* L.), peach (*Prunus persica* Batch.), nectarine (*Prunus persica* L. Batsch.), pear (*Pyrus communis* L.) and apple (*Malus domestica* Borth.) trees [14–21]. In particular, apples appear to be good targets for biofortifcation because they are widely consumed throughout the world and have high contents of dietary antioxidants that play an important role in human health [22]. The conducted experiments on apple trees have demonstrated that foliar sprays of I or Se were successful in enriching fruit with those micronutrients [14–16,19,21]. However, in these studies, combined spraying with I and Se were not tested despite that Prom-U-Thai et al. [23] concluded that from the economical point of view, combined sprays of several micronutrients (referred to sprays of "cocktail solutions") should be favoured. This spray strategy requires to perform field studies on the efficiency of the use of cocktail solutions in improving the utilisation of exogenously applied nutrients by plants, as well as on the potential of spray burn and its consequences.

Given that preharvest spray of calcium chloride ($CaCl_2$) at high rates is commonly applied in apple orchards, at least in Central and Eastern Europe, to improve the storability of mid- and late-season apples, it is justified to assess the efficiency of combined sprays of I, Se and $CaCl_2$. Therefore, the main purpose of this study was to examine the impacts of preharvest sprays of solutions containing I, Se and Ca, using different mineral species of I and Se, on enrichment of apples with those nutrients. The additional aims were to evaluate the effects of the above-mentioned sprays on the external and internal quality features of apples, and their storability.

## 2. Materials and methods

### 2.1 Plant material and growing conditions

The experiment was carried out as part of the statutory activity of the National Institute of Horticultural Research (NIHR), Skierniewice, Poland, approved by the Polish Ministry of Science and Higher Education (research topic 4.1.2). All permits were obtained from the owner of a commercial apple orchard where the study was conducted and from NIHR where most of the analyses were performed.

The study was conducted in 2020–2021 at 'Red Jonaprince' apple orchard in Central Poland (51˚ 48'N latitude and 20˚ 51'E longitude; 162 m above sea level) where the long-term averages of annual air temperature and precipitation are 5.5˚C and 525 mm, respectively.

Apple trees, grafted on M.9 rootstock, were planted in the spring of 2014 in brown soil (Cambisols), with the physicochemical properties given in Table 1. Soil parameters were determined based on a composite sample consisting of 20 subsamples taken in the early spring of 2020 across the entire experimental field from a depth of 0–30 cm from the surface of herbicide strips along tree rows. Except for total Se concentration, the analytical methods and apparatus used to determine soil parameters have been described in a previous paper by Wójcik and Wójcik [21]. Total Se in soil was determined by extraction with aqua regia, according to the procedure described by Dhillon et al. [24], using an inductively coupled plasma mass spectrometer (X-Series [II] ICP-MS; Thermo Fisher Scientific, Waltham, MA, USA).

The experimental trees grew in north–south oriented single rows of 100 m long each at a 3.5 m (between rows) × 1.2 m (within row) spacing (2381 plants ha$^{-1}$). Trees were trained as a spindle up to a height of 3.0 m by annual dormant pruning.

To control fruit load, foliar spray of Globaryll 100 SL (6-benzyladenine) at a rate of 1 L ha$^{-1}$ was performed annually when fruitlets were 10–12 mm in diameter (BBCH 71–72 according to the phenological scale of apple tree development proposed by Meier et al. [25]. Additionally, approximately 2 months before harvest (BBCH 75–77), small, scarred, blemished and/or deformed fruitlets were aborted by hand. No algae-based products were foliar applied in either season.

Tree protection against pathogens (mainly *Erwinia amylovora*, *Venturia inaequalis*, *Podosphaera leucotricha*, *Neonectria galligena*) and pests (such as *Cydria pomonella*, *Hoplocampa testudiea*, *Tetranychus urticae*, *Aphis pomi*, *Aculus schlechtendali*) was done according to up-to-date recommendations for integrated apple production using mostly selective compounds. Pesticides were used when an infection or infestation risk was identified. Control against pathogens and pests was performed from the bud breaking stage (BBCH 10) up to 4 weeks before harvest (BBCH 81). To control apple decay during storage, Captan 80 WDG fungicide (captan, Arysta Life Science, Nogueres, France) was used in the last spray treatment at a rate of 2 L ha$^{-1}$ as proposed by the manufacturer.

**Table 1. Soil characteristics of the layer of 0–30 cm prior to start of the study.**

| Parameter | Value | Nutrient abundance |
|---|---|---|
| The bulk density (g cm$^{-3}$) | 1.6 | - |
| Sand—0.05–2 mm (%) | 56 | - |
| Silt—0.002–0.05 mm (%) | 32 | - |
| Clay—< 0.002 mm (%) | 12 | - |
| Organic matter (g C kg$^{-1}$) | 18 | - |
| pH $_{(KCl)}$ | 6.6 | - |
| Total nitrogen (mg kg$^{-1}$) | 866 | medium |
| Available calcium* (mg kg$^{-1}$) | 720 | optimal |
| Available phosphorus* (mg kg$^{-1}$) | 44 | optimal |
| Available potassium* (mg kg$^{-1}$) | 102 | optimal |
| Available magnesium* (mg kg$^{-1}$) | 42 | optimal |
| Active oxides of iron (g kg$^{-1}$) | 3.6 | - |
| Active oxides of aluminum (g kg$^{-1}$) | 2.0 | - |
| Active oxides of manganese (mg kg$^{-1}$) | 16 | - |
| Total iodine (mg kg$^{-1}$) | 2.7 | - |
| Total selenium (mg kg$^{-1}$) | 0.21 | - |

* Determined in a solution of 0.03 N acetic acid containing ethylenediaminetetraacetic acid (EDTA).

Orchard floor management consisted of maintaining 1-m-wide herbicide-treated strips along the tree rows and mowed sod between the rows. The strips along the tree rows were maintained by the application of post-emergence herbicides of Roundup Active 360 SL (glyphosate, Monsanto Europe S.A., Antwerp, Belgium) in mid-May and late July at a rate of 5 L ha$^{-1}$ in each treatment and Agil-S 100 EC (propaquizafop, Adama Polska, Warsaw, Poland) in late August at a dose of 1 L ha$^{-1}$.

Trees were drip-irrigated to maintain soil moisture near field capacity to a depth of 30 cm. They were supplied with nitrogen (N) as ammonium nitrate ($NH_4NO_3$; 34% N; Anwil S.A., Włocławek, Poland) and Ca-nitrate [$Ca(NO_3)_2$; 15.5% N, 19% Ca; Yara Poland, Szczecin, Poland] and potassium (K) as K-sulphate ($K_2SO_4$; 42% K; K + S Kali GmbH, Kassel, Germany). The fertilisers were broadcast over the surface of the herbicide strips at annual rates of 60 kg N ha$^{-1}$ and 80 kg K ha$^{-1}$. The total N rate was divided into two equal parts. The first dose was applied at the swollen-breaking bud stage (BBCH 03–07) using $NH_4NO_3$, and the remaining amount was applied immediately after bloom (BBCH 67–69) as $Ca(NO_3)_2$. Fertiliser K was applied once, together with the first dose of fertiliser N. The rates of N and K and the timing of their application were in line with the recommendations by Wójcik [26] for mature high-density apple orchards grown in soil with moderate organic matter and optimal K availability.

## 2.2 The tested spray treatments and trial setup

Trees were sprayed with I, Se and Ca, 2 weeks before harvest—BBCH 81 (on 12 and 15 September 2020 and 2021, respectively). The tested spray solutions contained I as K-iodide (KI; 76.5% I) or K-iodate ($KIO_3$; 59.3% I), Se as Na-selenite ($Na_2SeO_3$; 45.7% Se) or Na- selenate ($Na_2SeO_4$; 41.8% Se), and Ca as $CaCl_2$ (28% Ca) at amounts of 0.5 kg I, 0.25 kg Se and 7 kg Ca per ha. The salts of I and Se were technical grade chemicals (Sigma-Aldrich, Poznań, Poland), whereas $CaCl_2$ was applied as a commercial fertiliser (CIECH, Warsaw, Poland). The rates of I, Se and Ca used in the spray treatments were high, considering those applied in apple orchards by Budke et al. [15] and Wójcik and Wójcik [21] for I, by Groth et al. [19] and Budke et al. [15] for Se, and by Wójcik [27] and Raese and Drake [28] for Ca.

To examine the effects of species of I and Se used in the spray solutions containing $CaCl_2$ on the uptake rate of exogenously applied Ca by apples, we tested also $CaCl_2$ spray performed at the same timing and dose as the combined spray treatments of I, Se and Ca. Control trees were not sprayed with either I, Se and Ca or water. The acronyms of the tested spray combinations and the solution concentrations used are given in Table 2.

**Table 2. Acronyms of the tested spray combinations and used solution concentrations in 'Red Jonaprince' apple orchar.**

| Combination | Acronym | Spray solution concentration (%) |
|---|---|---|
| Combined spray of potassium iodide, sodium selenite and calcium chloride | 'I-de + Se-ite + Ca' | 3.49 |
| Combined spray of potassium iodide, sodium selenate and calcium chloride | 'I-de + Se-ate + Ca' | 3.50 |
| Combined spray of potassium iodate, sodium selenite and calcium chloride | 'I-te + Se-ite + Ca' | 3.52 |
| Combined spray of potassium iodate, sodium selenate and calcium chloride | 'I-te + Se-ate + Ca' | 3.52 |
| Spray of calcium chloride | 'Ca' | 3.33 |
| No spray | 'Control' | - |

To improve wetting of fruit surfaces by the tested spray solutions, the non-ionic surfactant Tween® 20 (polyoxyethylene sorbitan monolaurate; Sigma-Aldrich, St. Louis, Missouri, USA) was added at a rate of 0.3 L ha$^{-1}$. The sprays were applied with a cross-flow orchard sprayer with an air deflector (Agrola Turbo 1000v Optimum; AGROLA, Platkowice, Poland) equipped with hollow-cone nozzles using 750 L of water per ha. The spray treatments were performed on both sides of the tree rows in the morning (09:00–11:00 h) when leaves/fruit were dry, the wind velocity was < 2 m s$^{-1}$, air temperature varied from 10 to 14˚C, and the relative air humidity (RH) was within the range of 75–89%.

Over the entire period of the study, the same trees were used for the given spray combinations. The experiment was set up in a completely randomised block design with four replications, with a replicate of each combination within one row. Each plot consisted of 10 trees. Plots within a row were separated by two trees. A buffer row was left untreated between every two adjacent sprayed rows to reduce contamination between treatments.

## 2.3 Measurements and observations

### 2.3.1 Leaf spray damage and cold injuries of buds and shoots.
Leaf injury, manifested as chlorosis/necrosis, was assessed 15 d after spray application on approximately 150 leaves from each plot located from the middle portion of current-season shoots on both sides of rows on the outside of the tree canopy at a height of 1.5–2.0 m above the ground. Leaf burn was assessed visually on a scale of 1 to 5, where 1 represents lack of leaf blade injury and 5 indicates chlorosis/necrosis > 75% of the leaf area.

Cold injuries of buds and shoots were assessed on 50 current-season shoots per plot, located identical to those for the determination of leaf spray burn. The shoots were excised at the stage of dormancy/beginning of leaf bud swelling (BBCH 00/01, on 4 March 2021 and 2022) and placed into 20-L containers in which the level of tap water allowed the shoots to be immersed to a height of about 5 cm. The shoots were kept for 14 d at room temperature. Then, buds were dissected vertically to evaluate their section colour. Brown discolouration or necrosis was scored as cold damage, while the green colour was classified as alive. Cold bud damage was expressed as the percentage of browned/necrotic buds.

Cold injury of shoots was determined according to the electrolyte leakage (EL) method described by Wójcik [27]. Briefly, six shoots randomly selected from each container (replication) were rinsed with deionised water, dried with a paper towel and cut into approximately 2 cm segments without buds. Then, 60 g of cut shoots were placed into a 200 mL glass beaker containing 40 mL of deionised water and incubated in a water bath (LWTc; WSL Świętochłowice, Poland) at room temperature for 24 h. The initial electrical conductivity (EC1) of the medium was measured using an electrical conductivity analyser (Orion Star A2/2; Thermo Fisher Scientific, Waltham, MA, USA). Subsequently, the samples were heated to 100˚C for 30 min to allow maximum leakage from shoot segments. After cooling to room temperature, the electrical conductivity was remeasured (EC2). The electrolyte leakage was calculated using the equation EL (%) = EC1/EC2 ×100.

### 2.3.2 Tree yielding and fruit quality at harvest.
The total fruit yield was determined separately for each plot. Harvest was performed once at commercial maturity (on 23 and 26 September 2020 and 2021, respectively) when the ethylene concentrations in the core cavity of the control fruit were within the range of 1.5–5.0 μL L$^{-1}$.

Fruit quality was assessed based on single fruit weight, russeting and skin blushing, flesh firmness (FF), soluble solid concentration (SSC), and titratable acidity (TA). Mean fruit weight, as well as russeting and skin blushing, were determined on an approximately 20 kg bulk sample per plot. Russeting and skin blushing were rated on a scale from 1 (no russeting/

blush) to 5 (russeting/blush on > 75% of the skin surface). Flesh firmness, SSC and TA of apples were measured on 30 fruit of similar size taken randomly from each of the 20 kg bulk samples per plot. Firmness was measured on two opposite peeled sides of each fruit using an EPT-1R penetrometer (Lake City Technical Products, Kelowna, BC, Canada) with an 11 mm diameter tip. The SSC was measured with an Atago PR-101 electronic refractometer (Atago Co., Ltd., Tokyo, Japan) using juice squeezed from the fruit homogenate at 20°C. Titratable acidity was determined by titrating the fruit homogenate with 0.1 $N$ sodium hydroxide to pH 8.1 using a Mettler Toledo DL 50 Graphix automatic titrator (Mettler-Toledo AG., Schwerzenbach, Switzerland). The results of the TA represent the malic acid content, expressed as a percentage.

**2.3.3 Iodine, Se and Ca in fruit at harvest.** Iodine, Se and Ca were determined in edible fruit parts (skin plus flesh) on 35 apples of similar size taken randomly from a 20 kg bulk sample from each plot. To remove any spray residue from the epicuticular wax layer, the fruit was rinsed with 0.01 M hydrochloric acid (HCl) and then with double-deionised water. After drying with a paper towel, two quarter-size pieces were cut from the opposite sides of each fruit and weighed by electronic balance (TP-3/1; FAWAG S.A., Lublin, Poland). Fruit samples were dried at 60°C in a forced-draft oven (UT 6760; Heraeus, Hanau, Germany) until a constant weight, reweighed and ground in a variable speed rotor mill (MF 10/2; IKA Poland, Warsaw, Poland) to pass through a 32-mesh screen.

Iodine was determined according to the procedure recommended by the Polish Committee for Standardization [29]. Briefly, 0.5 g of ground fruit sample, 15 mL of distilled water and 1 mL of 25% tetra-methylammonium hydroxide (TMAH) were mixed in a 30 mL Falcon tube and heated at 90°C for 3 h. Then, the tubes were filled with double-deionised water, mixed, spun down at 4500 rpm for 15 min and filtered through a 0.24 μm membrane (MilliporeSigma, Burlington, MA, USA). Iodine was measured with an inductively coupled plasma mass spectrometer (X-Series [II] ICP-MS; Thermo Fisher Scientific, Waltham, MA, USA).

Selenium and Ca in fruit samples were determined according to the procedure set by the Polish Committee for Standardization [30], with a partial modification proposed by Wójcik and Borowik [31]. Briefly, 0.5 g of ground fruit sample was transferred to a Teflon reaction vessel and 5 mL of 65% nitric acid ($HNO_3$) and 2.5 mL of 30% hydrogen peroxide ($H_2O_2$) were added. This mixture was kept overnight and then microwave-digested (MLS 1200, Milistone, Inc., Monroe, CT, USA) using the following temperature programme: 130°C for 5 min, 160°C for 10 min and 200°C for 20 min. After finishing the reaction, the vessels were cooled and placed into a water bath at 80°C for 30 min. The sample solutions were transferred quantitatively to 25 mL volumetric flasks that were subsequently filled with double-deionised water. After filtrating the solution through a 0.24 μm membrane, Se and Ca were determined by ICP-MS.

To control the measurement accuracy of I, Se and Ca in the studied fruit samples, two Standard Reference Materials [SRM 1515: apple leaf tissues with concentrations of 1.52% Ca and 0.30 mg I kg$^{-1}$ of dry weight (DW) and SRM 1573a: tomato leaves containing 5.05% Ca, 0.85 mg I kg$^{-1}$ and 0.054 mg Se kg$^{-1}$ DW] were used. The fruit concentrations of I, Se and Ca were expressed on a fresh weight (FW) basis.

**2.3.4 Fruit storability.** Immediately after harvest, the apples were transported to refrigerated air storage where they were held for 150 d at 1°C and RH of 90–95%. After removing fruit from cold storage, they were kept for 7 d at 20°C and 60–70% RH to simulate the shelf life period.

Apple storability was assessed based on FF, SSC, TA, the weight loss of fruit during the storage and shelf life periods, and the incidence of physiological disorders and fungal diseases.

Firmness, SSC and TA of fruit were determined on 30 apples from each plot according to the same procedures as at harvest.

Fruit disorders and diseases were visually evaluated after the shelf life period on a 20 kg bulk sample from each plot. To assess disorders/diseases developed in the inner cortex, each fruit was cut longitudinally into four parts. The number of fruit affected by each disorder or disease was recorded. The results were expressed as a percentage share of affected fruit compared to the total number of fruit in the bulk sample.

Since Ca ions do not tend to volatilise, only the loss rates of I and Se from stored apples were determined. The losses were calculated by subtracting from amounts I or Se in edible parts of fruit at harvest and their amounts after the shelf life period. Fruit I and Se after storage were determined on 35 apples from the plot in the same manner as at harvest.

### 2.4 Statistical analysis

All data were subjected to a one-way analysis of variance. The data of cold bud injury, and the incidence of disorders and diseases were transformed according to y = arc sin x. The values of the above-mentioned parameters for each combination in the given Tables 3 and 7 were back-transformed. In the Tables 3–7, the means ± standard deviations of the four replicates for each combination are shown. The analyses were performed separately for each growing season using Tukey's multiple range test at P ≤ 0.05 by means of Statistica 10 software (StatSoft Polska, Krakow, Poland).

## 3. Results

### 3.1 Spray phytotoxicity, cold injuries, and yield

Leaves from the control trees had no signs of spray injury (Table 3). However, all spray combinations caused leaf damage up to 25% of the blade area. The intensity of leaf phytotoxicity did not depend on the spray treatments tested (Table 3).

The EL values of the shoot tissues did not differ between the combinations (Table 3). Cold bud damage was slight and not affected by the spray combinations (Table 3).

Across the tested combinations, fruit yields averaged 20.7 kg tree$^{-1}$ in 2020 and 20.4 kg tree$^{-1}$ in 2021 (corresponding to 49.3 tons ha$^{-1}$ and 48.6 tons ha$^{-1}$, respectively), which can be believed as moderate for high-density apple orchards under Polish conditions. The spray treatments did not affect fruit yield (Table 3).

### 3.2 Fruit quality and fruit nutrient contents at harvest

None of the spray treatments affected the mean fruit weight, fruit skin russeting (mainly around the calyx) or apple colouring (Table 4).

**Table 3. Effects of combined sprays of iodine (I), selenium (Se) and calcium (Ca) on phytotoxicity, cold injuries and yield of 'Red Jonaprince' apple trees.**

| Spray treatment | Leaf injury (1–5) | | Electrolyte leakage of shoots (%) | | Bud injury (%) | | Fruit yield (kg tree$^{-1}$) | |
|---|---|---|---|---|---|---|---|---|
| | 2020 | 2021 | 2021 | 2022 | 2021 | 2022 | 2020 | 2021 |
| I-de + Se-ite + Ca | 1.9 ± 0.2 a | 1.8 ± 0.1 a | 5.2 ± 2.1 a | 5.4 ± 1.2 a | 3.0 ± 0.6 a | 3.3 ± 1.3 a | 20.0 ± 0.7 a | 20.0 ± 0.5 a |
| I-de + Se-ate + Ca | 1.8 ± 0.1 a | 1.9 ± 0.2 a | 5.7 ± 1.5 a | 4.6 ± 2.1 a | 3.9 ± 1.1 a | 3.4 ± 1.0 a | 20.4 ± 0.5 a | 20.4 ± 0.4 a |
| I-te + Se-ite + Ca | 1.8 ± 0.3 a | 1.9 ± 0.2 a | 6.4 ± 1.9 a | 6.0 ± 2.7 a | 2.7 ± 0.8 a | 3.2 ± 1.0 a | 19.9 ± 0.8 a | 19.7 ± 0.7 a |
| I-te + Se-ate + Ca | 1.9 ± 0.1 a | 1.8 ± 0.1 a | 4.4 ± 1.5 a | 4.9 ± 1.6 a | 3.5 ± 1.4 a | 3.2 ± 1.7 a | 21.6 ± 1.0 a | 21.1 ± 1.0 a |
| Ca | 1.7 ± 0.1 a | 1.9 ± 0.1 a | 4.9 ± 2.1 a | 4.6 ± 2.0 a | 3.4 ± 1.3 a | 3.6 ± 1.4 a | 20.8 ± 0.5 a | 20.8 ± 0.7 a |
| Control | 1.0 ± 0 b | 1.0 ± 0 b | 5.3 ± 2.2 a | 7.2 ± 3.0 a | 3.7 ± 1.6 a | 3.7 ± 1.2 a | 21.5 ± 1.0 a | 20.2 ± 0.7 a |

Means with the same letter within each column are not significantly different by Tukey's multiple range test at α = 0.05.

**Table 4. Effects of combined sprays of iodine (I), selenium (Se) and calcium (Ca) on 'Red Jonaprince' apple quality at harvest.**

| Spray treatment | Mean fruit weight (g) | | Skin russeting (1–5) | | Fruit blushing (1–5) | | Soluble solid concentration (%) | | Firmness (N) | | Titratable acidity (%) | |
|---|---|---|---|---|---|---|---|---|---|---|---|---|
| | 2020 | 2021 | 2020 | 2021 | 2020 | 2021 | 2020 | 2021 | 2020 | 2021 | 2020 | 2021 |
| I-de + Se-ite + Ca | 180 ± 4 a | 177 ± 4 a | 1.4 ± 0.2 a | 1.4 ± 0.3 a | 4.4 ± 0.3 a | 4.1 ± 0.1 a | 13.8 ± 0.1 a | 14.0 ± 0.2 a | 84 ± 1 a | 83 ± 1 a | 0.89 ± 0.04 a | 0.90 ± 0.03 a |
| I-de + Se-ate + Ca | 181 ± 5 a | 177 ± 5 a | 1.6 ± 0.3 a | 1.3 ± 0.2 a | 4.3 ± 0.2 a | 4.2 ± 0.1 a | 13.9 ± 0.1 a | 14.1 ± 0.2 a | 82 ± 2 a | 82 ± 2 a | 0.91 ± 0.07 a | 0.91 ± 0.05 a |
| I-te + Se-ite + Ca | 179 ± 5 a | 178 ± 3 a | 1.5 ± 0.2 a | 1.5 ± 0.2 a | 4.3 ± 0.3 a | 4.2 ± 0.2 a | 13.9 ± 0.1 a | 14.0 ± 0.2 a | 83 ± 1 a | 84 ± 1 a | 0.94 ± 0.03 a | 0.89 ± 0.04 a |
| I-te + Se-ate + Ca | 180 ± 3 a | 178 ± 3 a | 1.2 ± 0.3 a | 1.5 ± 0.2 a | 4.2 ± 0.1 a | 4.2 ± 0.2 a | 13.8 ± 0.2 a | 14.1 ± 0.1 a | 82 ± 2 a | 83 ± 1 a | 0.90 ± 0.02 a | 0.92 ± 0.03 a |
| Ca | 181 ± 3 a | 175 ± 3 a | 1.4 ± 0.2 a | 1.3 ± 0.2 a | 4.3 ± 0.1 a | 4.2 ± 0.1 a | 13.8 ± 0.1 a | 14.2 ± 0.1 a | 84 ± 1 a | 84 ± 1 a | 0.94 ± 0.03 a | 0.90 ± 0.01 a |
| Control | 178 ± 3 a | 175 ± 3 a | 1.4 ± 0.2 a | 1.4 ± 0.2 a | 4.2 ± 0.2 a | 4.2 ± 0.2 a | 14.0 ± 0.2 a | 14.2 ± 0.1 a | 83 ± 1 a | 83 ± 1 a | 0.92 ± 0.05 a | 0.89 ± 0.02 a |

Means with the same letter within each column are not significantly different by Tukey's multiple range test at α = 0.05.

Flesh firmness of apples did not differ between the combinations, averaging 83 N in both seasons (Table 4). The contents of soluble solids and organic acids in the fruit were not affected by the spray treatments (Table 4).

Apples sprayed with solutions containing I, Se and Ca had more those nutrients than fruit from the control trees; the efficiencies of the above spray treatments in increasing fruit concentrations of I, Se and Ca were comparable (Table 5). Compared to the control fruit, concentrations of I, Se and Ca in apples sprayed with those nutrients were increased by 5442% and 4575% for I, by 4875% and 4400% for Se, and by 30% and 32% for Ca, on average, in 2020 and 2021, respectively. Regardless of this fact, apple Ca concentrations from trees sprayed only with Ca were comparable to fruit treated with I, Se and Ca (Table 5).

### 3.3 Fruit storability

After storage, the lowest FF and TA of apples occurred on the control plots (Table 6). Higher firmness and acidity of fruit were found for the Ca combination (Table 6). Regardless of the species of I and Se, apples sprayed with I, Se and Ca were the firmest and contained the most organic acids (Table 6). In both growing seasons, none of the tested spray treatments affected SSC in apples (Table 6).

Among physiological disorders, only bitter pit (BP) and internal breakdown (IB) were observed. The apples most affected by BP and IB were found in the control plots (Table 7). Combined sprays of I, Se and Ca, as well as the Ca treatment, reduced the incidence of BP and IB. Except for IB in 2020, the application of cocktail solutions had comparable impacts on the reduction of BP and IB as the Ca combination (Table 7). In 2020, apples sprayed only with Ca were more susceptible to IB than those treated with I, Se and Ca (Table 7).

**Table 5. Effects of combined sprays of iodine (I), selenium (Se) and calcium (Ca) on the amounts of those nutrients in edible parts of 'Red Jonaprince' apples at harvest.**

| Spray treatment | Fruit I (µg 100 g⁻¹ FW) | | Fruit Se (µg 100 g⁻¹ FW) | | Fruit Ca (mg 100 g⁻¹ FW) | |
|---|---|---|---|---|---|---|
| | 2020 | 2021 | 2020 | 2021 | 2020 | 2021 |
| I-de + Se-ite + Ca | 75.5 ± 5.2 a | 75.0 ± 4.1 a | 10.1 ± 1.5 a | 9.3 ± 0.8 a | 4.8 ± 0.2 a | 4.9 ± 0.3 a |
| I-de + Se-ate + Ca | 77.2 ± 3.3 a | 73.5 ± 5.1 a | 9.8 ± 1.4 a | 8.7 ± 1.4 a | 4.7 ± 0.3 a | 5.0 ± 0.2 a |
| I-te + Se-ite + Ca | 78.2 ± 2.2 a | 73.5 ± 5.1 a | 9.8 ± 0.9 a | 8.9 ± 1.0 a | 4.8 ± 0.2 a | 4.9 ± 0.2 a |
| I-te + Se-ate + Ca | 79.7 ± 4.3 a | 77.2 ± 5.6 a | 10.1 ± 0.8 a | 9.3 ± 1.1 a | 5.0 ± 0.2 a | 4.9 ± 0.2 a |
| Ca | 1.1 ± 0.3 b | 1.6 ± 0.4 b | 0.2 ± 0.1 b | 0.2 ± 0 b | 4.8 ± 0.2 a | 4.9 ± 0.2 a |
| Control | 1.4 ± 0.2 b | 1.6 ± 0.4 b | 0.2 ± 0 b | 0.2 ± 0 b | 3.7 ± 0.1 b | 3.7 ± 0.1 b |

Means with the same letter within each column are not significantly different by Tukey's multiple range test at α = 0.05.

**Table 6. Effects of combined sprays of iodine (I), selenium (Se) and calcium (Ca) on quality features of 'Red Jonaprince' apples after storage.**

| Spray treatment | Firmness (N) | | Soluble solid concentration (%) | | Titratable acidity (%) | |
|---|---|---|---|---|---|---|
| | 2020 | 2021 | 2020 | 2021 | 2020 | 2021 |
| I-de + Se-ite + Ca | 61 ± 2 a | 60 ± 1 a | 13.5 ± 0.1 a | 13.7 ± 0.2 a | 0.53 ± 0.02 a | 0.56 ± 0.02 a |
| I-de + Se-ate + Ca | 59 ± 2 a | 59 ± 1 a | 13.7 ± 0.2 a | 13.7 ± 0.2 a | 0.54 ± 0.02 a | 0.55 ± 0.03 a |
| I-te + Se-ite + Ca | 60 ± 2 a | 59 ± 2 a | 13.7 ± 0.2 a | 13.6 ± 0.1 a | 0.53 ± 0.03 a | 0.55 ± 0.02 a |
| I-te + Se-ate + Ca | 59 ± 2 a | 60 ± 2 a | 13.5 ± 0.1 a | 13.7 ± 0.2 a | 0.51 ± 0.02 a | 0.55 ± 0.02 a |
| Ca | 53 ± 1 b | 52 ± 1 b | 13.7 ± 0.2 a | 13.6 ± 0.2 a | 0.46 ± 0.01 b | 0.48 ± 0.01 b |
| Control | 47 ± 2 c | 45 ± 1 c | 13.6 ± 0.2 a | 13.6 ± 0.1 a | 0.37 ± 0.02 c | 0.41 ± 0.01 c |

Means with the same letter within each column are not significantly different by Tukey's multiple range test at α = 0.05.

After storage, apples were infected by *Neofabraea* spp., *Botrytis cinerea* and *Penicillium expansum*, causing bull's eye rot, grey mould, and blue mould, respectively. In the case of bull's eye rot, combined sprays of I, Se and Ca eliminated this disease, while the Ca treatment reduced its occurrence only in 2020 (Table 7). The incidence of grey mould and blue mould was slight and was not affected by the spray combinations (Table 7).

## 4. Discussion

### 4.1 Tree injuries and fruit quality at harvest

Leaf injuries of perennial fruit crops may decrease their cold hardiness, and fruit yield and quality [32]. These responses result from a limited supply of assimilates from leaves to other parts of trees. In our study, combined application of I, Se and Ca and application of only Ca caused considerable leaf burn (Table 3). Simultaneously, the above spray treatments had no effect on cold injury of buds and current season shoots and on fruit yield. Similarly, Wójcik [27] demonstrated that Ca sprays applied one week before harvest (the first week of October) did not affect the cold hardiness of 'Jonagold' apple trees and fruit yield despite leaf burn and even defoliation. Thus, the lack of impacts of the tested spray treatments in the present experiment on the cold hardiness of trees and fruit yield probably resulted from the fact that defoliation might occur in the period of natural leaf fall. The lack of responses of the tested spray treatments to cold injury of buds and shoots and fruit production could also result from relatively high air temperatures in the winter period. The lowest air temperatures at the site of our study were −16˚C in 2020/2021 (on 18 January) and -9˚C in 2021/2022 (on 11 March), while the critical temperature for most apple varieties grown in Poland in the dormancy periods is −20˚C [33].

**Table 7. Effects of combined sprays of iodine (I), selenium (Se) and calcium (Ca) on the incidence of physiological disorders and diseases of 'Red Jonaprince' apples after storage.**

| Spray treatment | Bitter pit (%) | | Internal breakdown (%) | | Bull's eye rot (%) | | Gray mold (%) | | Blue mold (%) | |
|---|---|---|---|---|---|---|---|---|---|---|
| | 2020 | 2021 | 2020 | 2021 | 2020 | 2021 | 2020 | 2021 | 2020 | 2021 |
| I-de + Se-ite + Ca | 0.5 ± 0.2 b | 0.6 ± 0.1 b | 0.2 ± 0.1 c | 0.3 ± 0.1 b | 0 ± 0 c | 0 ± 0 b | 0 ± 0 a | 0.2 ± 0.1 a | 0.3 ± 0.1 a | 0 ± 0 a |
| I-de + Se-ate + Ca | 0.7 ± 0.2 b | 0.7 ± 0.2 b | 0.2 ± 0.1 c | 0.5 ± 0.2 b | 0 ± 0 c | 0 ± 0 b | 0 ± 0 a | 0.4 ± 0.2 a | 0.4 ± 0.2 a | 0 ± 0 a |
| I-te + Se-ite + Ca | 0.4 ± 0.1 b | 0.7 ± 0.4 b | 0.2 ± 0.1 c | 0.5 ± 0.2 b | 0 ± 0 c | 0 ± 0 b | 0 ± 0 a | 0.4 ± 0.2 a | 0.3 ± 0.2 a | 0 ± 0 a |
| I-te + Se-ate + Ca | 0.6 ± 0.3 b | 0.8 ± 0.4 b | 0.2 ± 0.2 c | 0.6 ± 0.2 b | 0 ± 0 c | 0 ± 0 b | 0 ± 0 a | 0.3 ± 0.2 a | 0.2 ± 0.1 a | 0 ± 0 a |
| Ca | 0.5 ± 0.3 b | 0.5 ± 0.2 b | 0.9 ± 0.2 b | 0.4 ± 0.1 b | 0.8 ± 0.2 b | 1.4 ± 0.2 a | 0 ± 0 a | 0.4 ± 0.2 a | 0.2 ± 0.1 a | 0 ± 0 a |
| Control | 3.7 ± 0.4 a | 2.9 ± 0.9 a | 2.2 ± 0.2 a | 1.6 ± 0.3 a | 2.6 ± 0.3 a | 1.6 ± 0.3 a | 0 ± 0 a | 0.5 ± 0.3 a | 0.3 ± 0.1 a | 0 ± 0 a |

Means with the same letter within each column are not significantly different by Tukey's multiple range test at α = 0.05.

It is worth noting that the intensity of leaf burn in trees sprayed with Ca and with solutions containing I, Se and Ca were comparable (Table 3), suggesting that only $CaCl_2$ use caused leaf injury. However, Wójcik and Wójcik [21] demonstrated that the preharvest spray of I used at the same dose and on the same apple variety as in the present study, resulted in leaf injury up to 28% of their area. Budke et al. [15] also found that the preharvest application of I at a rate of 0.25 kg ha$^{-1}$ per metre canopy height of 'Jonagold' apple trees (which corresponded to a dose of about 0.5 kg I ha$^{-1}$) led to leaf injury. Therefore, we suggest that the presence of $CaCl_2$ in the cocktail solutions mitigated/neutralised the phytotoxicity potential of I. This suggestion seems to be justified because $CaCl_2$ is a well-known humectant. Its value of the point of deliquescence (POD), defined as RH at which the liquefaction of a salt into a solute takes place, is as low as 30% [34]. If, in the period of the tested sprayings, RH was 75–89%, thus the presence of $CaCl_2$ in the cocktail solutions could prolong their liquefaction on the plant surface, thereby reducing the risk of leaf burn.

Apple appearance, size and some internal quality features, such as firmness and contents of soluble solids and organic acids, are crucial in terms of consumer preference [35]. In the present study, none of the tested spray treatments caused apple skin russeting despite leaf injury (Tables 3 and 4). Wójcik and Wójcik [21] also found no apple skin damage because of preharvest I sprays, despite observed leaf burn, indicating that the apple fruit cuticle is more resistant to the spray solutions than the leaf surface. However, it is not surprising because the fruit epidermis of many deciduous tree species is covered by a thicker wax layer than the leaf surface, consequently increasing the protection of fruit skin cells against environmental stress [36].

In our experiment, apple size, firmness and contents of soluble solids and acids were comparable among the tested combinations, suggesting that leaf photosynthetic capacity and assimilate transport rates from leaves to apple tissues were not dramatically reduced. Wójcik et al. [37,38] also failed to change the internal quality of 'Granny Smith' apples and 'Conference' (*Pyrus communis* L.) pears, respectively, through preharvest Ca sprays, despite leaves being damaged. On the second hand, Wójcik and Wójcik [21] demonstrated that preharvest I spray at a rate of 1.5 kg ha$^{-1}$ simultaneously led to leaf burn and decreased SSC in apples at harvest. Moreover, Budke et al. [15] increased SSC in 'Fuji' apples because of combined sprays of $KIO_3$ and $KNO_3$ despite leaf burn. Thus, we suggest that apple quality responses to preharvest sprays of nutrients under conditions of leaf burn depend on the nutrient/salt used, the intensity of leaf injury, plant development stage, variety and weather factors occurring immediately after spraying, such as air temperature, rainfall and radiation.

## 4.2 Iodine, Se and Ca in fruit at harvest

In the present experiment, 'Red Jonaprince' apples from the control trees contained < 2 μg I 100 g$^{-1}$ FW (Table 5). Budke et al. [15,16] demonstrated that native I concentrations in 'Jonagold' and 'Fuji' apples were ≤ 1.5 μg 100 g$^{-1}$ FW, whereas in 'Golden Delicious' apples, they were only 0.4 μg I 100 g$^{1}$ FW. Thus, apples can be considered as poor I sources, which probably results from the reduced phloem mobility of this micronutrient [16].

Compared to the control apples, fruit sprayed with the cocktail solutions contained about 50 times more I (Table 5). Given that (i) the recommended dietary intake (RDI) of I for adults is 150 μg d$^{-1}$ [39], (ii) the mean apple weight across the spray combinations with the use of the cocktail solutions was 180 g and 177 g in 2020 and 2021, respectively, (iii) fruit I concentrations from those combinations averaged 78 μg 100 g$^{-1}$ FW and 75 μg 100 g$^{-1}$ FW in 2020 and 2021, respectively, and (iv) inedible parts of an apple (core, seeds and stem) is approximately 10% of its mass as demonstrated by Wójcik [40], thus the intake of a single apple covered an RDI in 84% in 2020 and in 79% in 2021.

Only a few reports have examined the absorption rate of different I species by apples. Budke et al. [15] demonstrated that KI sprays at rates of 0.25–2.5 kg I ha$^{-1}$ per metre canopy height (which corresponded to about 0.5–5 kg I ha$^{-1}$) were more effective in improving 'Jonagold' apple I concentration than those with the use of KIO$_3$. Wójcik and Wójcik [21] also found that preharvest KI sprays at a rate of 1 kg or 1.5 kg I per ha enhanced the I concentration to a greater degree in 'Red Jonaprince' apples than KIO$_3$ sprays. In the present experiment, regardless of the I species used, apples sprayed with the cocktail solution had comparable I concentrations (Table 5). The lack of differences in the absorption rate of I from KI and KIO$_3$ could result from lowering the POD of the cocktail solutions to a similar value by the presence of CaCl$_2$. Lawson et al. [41] demonstrated that the addition of CaCl$_2$-based fertiliser to an aqueous spray solution containing I increased the concentration of this micronutrient in edible parts of butterhead lettuce (*Lactuca sativa*). The positive effect of CaCl$_2$ on the absorption rate of mineral I species by epidermal cells is particularly important for KIO$_3$ because the POD value of this salt at 20˚C is as high as 93.8% [42]. Thus, if the RH is below this POD value, KIO$_3$ will be crystalised on the fruit surface, thereby reducing absorption rate of I ions.

In the case of Se, its concentration in apples from the control trees was 0.2 µg 100 g$^{-1}$ FW in both seasons (Table 5). In Germany, Groth et al. [19] found that apple Se concentrations for several varieties varied from 0.1 to 0.7 µg 100 g$^{-1}$ FW. Similar apple Se concentrations have also been recorded in Greece by Pappa et al. [43], in Slovakia by Kadrabowa et al. [44] and in Croatia by Klapec et al. [45].

According to Gupta and Gupta [12] and Lyons [46], foliar Se sprays are highly effective in enriching plants with this nutrient, at least for those with edible aerial parts above the ground. Our results confirmed this opinion because the sprays of the cocktail solutions enhanced apple Se amounts by about 50-fold (up to 10.1 µg 100 g$^{-1}$ FW) (Table 5). In the experiment by Groth et al. [19], combined application of Se with Ca (as Wuxal$^®$ Ascofol Ca fertiliser) at rates of 0.075–0.15 kg Se ha$^{-1}$ per metre canopy height (which corresponded to approximately 0.15–0.30 kg Se ha$^{-1}$) increased apple Se concentrations from 3.1 to 13.9 µg 100 g$^{-1}$ FW. Similar to our study, no significant differences in the absorption rate of exogenously applied Se between selenate and selenite were found in apples. In contrast, Lyons [46] showed that foliar sprays of selenate were more effective in improving Se in plants than selenite species. Comparable effects of sprays of cocktail solutions containing selenate or selenite on Se absorption rate by apples obtained in our study could be caused by modifications of physico-chemical properties of the spray solutions by the presence of CaCl$_2$. Further investigations are needed to better understand the interactions between Ca salts and different Se species in spray solutions on the processes of Se absorption by apples.

The RDI of Se for adults is 55 µg d$^{-1}$ [39]. Given the mean apple weight from the plots sprayed with the cocktail solutions, Se concentrations in apples from the above plots (9.9 µg 100 g$^1$ FW in 2020 and 9.1 µg 100 g$^{-1}$ FW in 2021) and the percentage contribution of inedible parts of an apple to its total mass, thus the Se amounts in an apple enriched with this micronutrient covered an RDI in 29% and 26% in 2020 and 2021, respectively. However, it is worth noting that the recorded ratios of I:Se in fruit sprayed with cocktail solutions (4.7–5.3) were within the optimal range of 4.4–8.8 in the human diet recommended by Smoleń et al. [47].

In the case of Ca, the RDI for adults is 800 mg d$^{-1}$ [39]. Considering Ca concentrations in apples sprayed with the cocktail solutions (4.8 mg 100 g$^{-1}$ FW in 2020 and 4.9 mg 100 g$^{-1}$ FW, on average), we can state that the Ca amounts in those fruit covered RDI in ca. 1%. Therefore, apples cannot be considered a target crop for Ca biofortification.

It is worth emphasizing that the efficiencies of sprays of the cocktail solutions and only Ca in improving fruit Ca concentration were comparable (Table 5), indicating that the absorption

rate of Ca from $CaCl_2$ by apples did not depend on the presence of I and Se in the spray solutions.

## 4.3 Fruit storability

Flesh firmness and acidity of apples are commonly utilised to assess their ripening [48]. In our study, apples sprayed with cocktail solutions, as well as sprayed only with Ca, were firmer and contained more organic acids after storage than the control fruit (Table 6), indicating that they were less ripe. The inhibited ripening process of the above-mentioned apples can be attributed to enhanced Ca concentrations in their tissues, since it has been well documented that Ca plays a critical role in the respiration process, ethylene production and consequently softening and fruit senescence [49].

It is worth noting that the FF and TA of apples sprayed with cocktail solutions were greater than fruit treated only with Ca (Table 6), despite comparable apple Ca concentrations (Table 5). We suggest that the increased apple Se concentrations through cocktail solution application might inhibit apple ripening processes. This seems to be true because Zhu et al. [50] proved that preharvest Se sprays decreased ethylene production and respiration rates in stored tomato (*Solanum Lycopersicon*) fruit. Babalar et al. [14] demonstrated that Se-sprayed apples evolved less ethylene during storage, which lead to decreases in FF and TA losses. Lower FF losses in pear and peach fruit during storage because of preharvest Se sprays has also been observed by Pezzarossa et al. [20]. Reduced ethylene production by Se-enriched apples may result from (i) an increased rate of Se-methionine formation, which decreases the amount of free methionine essential in the ethylene biosynthesis pathway or (ii) improved cell membrane integrity by increased antioxidant system activity [51,52]. However, Iqbal et al. [53] found in wheat (*Triticum* L.) that Se was able to inhibit the activity of 1-aminocyclopropane-1-carboxylate synthase, which is a key enzyme in ethylene biosynthesis. Further research is needed to examine both the action of Se in ethylene biosynthesis and its role in the processes of ripening and fruit senescence.

Bitter pit is a physiological disorder that occurs in many apple-growing regions. It is believed that the main factor causing BP is Ca deficiency in apple flesh tissues [54]. Therefore, summer Ca sprays are most often recommended in apple orchards to control BP [55,56]. Similarly, as in previous studies by Wójcik [27] and Wójcik et al. [37], we found that fall sprays of solutions containing Ca at a high rate were successful in reducing the incidence of BP (Table 7). Given that apples with increased Ca concentrations were less susceptible to BP (Tables 5 and 7), our results confirmed the opinion that BP is related to low Ca status in their tissues.

Another apple disorder recorded in our study was IB, which is associated with the senescence processes of fruit [57]. Our results agree with this because firmer apples with an increased organic acid content from trees sprayed with cocktail solutions or only with Ca were less susceptible to IB (Tables 6 and 7).

In many countries, including Poland, bull's eye rot of apples caused by *Neofabraea* spp. is the most serious postharvest disease [58,59]. In our study, bull's eye rot was also the main fungal disease occurring on the control apples (Table 7). However, sprays of I, Se and Ca completely eliminated this disease (Table 7). Given that in the first season, the Ca spray decreased bull's eye decay to a lesser extent than sprays of the cocktail solutions despite comparable apple Ca concentrations (Table 5), thus both Ca and Se or I used in the cocktail solutions were able to reduce the infection of apples by *Neofabraea* spp. As there is no evidence indicating that foliar I sprays can protect plants against fungal pathogens, we suggest that using Se in cocktail solutions might decrease apple decay by *Neofabraea* spp. Wu et al. [60,61]

demonstrated that Se treatments reduced postharvest diseases caused by *Botrytis cinerea* and *Penicilium expansum* on some fruit and vegetables. Moreover, Hanson et al. [62] demonstrated that high Se concentrations in leaf tissues of Indian mustard (*Brassica juncea*) decreased infections by *Alternaria brassiciola* and *Fusarium* spp. According to Wu et al. [60] and Somalraju et al. [63], the inhibitory effect of Se against fungal diseases is related to (i) the induction of reactive oxygen species in spores and mycelium, leading to the reduction in spore germination, mycelium growth and sporulation and/or (ii) increased production of phenolic secondary metabolites in plant tissues by eliciting the expression of genes in the phenylpropanoid pathway.

## 5. Conclusions

The results obtained of this study demonstrated that combined sprays of I (KI or KIO$_3$), Se (Na$_2$SeO$_3$ or Na$_2$SeO$_4$) and CaCl$_2$ applied at 2 weeks before harvest of 'Red Jonaprince' apples at rates of 0.5 kg I, 0.25 kg Se and 7 kg Ca ha$^{-1}$ were highly effective in enriching fruit with I and Se because their amounts in edible parts of an apple were increased about 50-fold. Calcium amounts in apples sprayed with cocktail solutions increased by only about 30%. However, the use of CaCl$_2$ with I and Se in the spray solutions probably mitigated phytotoxicity caused by I and Se, as well as improved apple storability, including reducing the incidence of BP.

Given the above information, as well as the fact that the tested sprays of the cocktail solutions did not affect apple appearance and their internal features at harvest, we conclude that preharvest fall spraying with solutions containing I, Se and Ca at high doses can be recommended in apple orchards, at least for mid- and late-season varieties sensitive to Ca-related disorders, to effectively improve the I and Se contents in apples and their storability.

## Author Contributions

**Conceptualization:** Paweł Wójcik.

**Formal analysis:** Paweł Wójcik.

**Investigation:** Paweł Wójcik.

**Methodology:** Paweł Wójcik.

**Resources:** Paweł Wójcik.

**Supervision:** Paweł Wójcik.

**Visualization:** Paweł Wójcik.

**Writing – original draft:** Paweł Wójcik.

**Writing – review & editing:** Paweł Wójcik.

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
