## [Decision Letter · Decision Letter 0]

14 Feb 2023

PONE-D-22-35045Effects of preharvest sprays of iodine, selenium and calcium on apple biofortification and their quality and storabilityPLOS ONE

Dear Dr. Wojcik,

Thank you for submitting your manuscript to PLOS ONE. After careful consideration, we feel that it has merit but does not fully meet PLOS ONE’s publication criteria as it currently stands. Therefore, we invite you to submit a revised version of the manuscript that addresses the points raised during the review process.

We look forward to receiving your revised manuscript.

Kind regards,

Hasan Sardar, Ph.D.

Academic Editor

PLOS ONE

Journal Requirements:

Reviewers' comments:

Reviewer's Responses to Questions

**Comments to the Author**

1. Is the manuscript technically sound, and do the data support the conclusions?

Reviewer #1: Yes

Reviewer #2: Yes

2. Has the statistical analysis been performed appropriately and rigorously? 

Reviewer #1: Yes

Reviewer #2: Yes

3. Have the authors made all data underlying the findings in their manuscript fully available?

Reviewer #1: Yes

Reviewer #2: Yes

4. Is the manuscript presented in an intelligible fashion and written in standard English?

Reviewer #1: Yes

Reviewer #2: Yes

5. Review Comments to the Author

Reviewer #1: The manuscript presents an industrial progress on the quality enhancement of apples. I have following concerns to further improve its quality.

What time of the day the treatments were applied?

Authors are suggested to provide an overview picture showing their treatments and expected results, that would be easy for a variety of readers to understand their key findings

Line 26; remove "with those nutrients"

Reviewer #2: This manuscript focuses on examining the effects of iodine (I), selenium (Se) and calcium (Ca) on the enrichment of ‘Red Jonaprince’ (Malus domestica Borth.) apples as well as fruit quality and their storability. Although the research topic and the approach are appropriate, but minor correction are necessary before acceptance of manuscript. Following are my suggestions

In introduction section, introduction of nutrients should be explained with detail and with latest examples of different fruit crops

Line 111. What type of pesticides and herbicides were used and at what time?

Line 141. Is there anything like water applied to the control plants?

In the results section explain the results of all treatments

Discussion is corroborative. The authors should highlight the reason of their result findings in the light of available literature.

6. PLOS authors have the option to publish the peer review history of their article (what does this mean?). If published, this will include your full peer review and any attached files.

Reviewer #1: No

Reviewer #2: No

---

## [Author Response · Author response to Decision Letter 0]

23 Feb 2023

According to the requirements of PLOS ONE:

- we have added information in the Method section about permits received related to conducting field study and performing analyses/measurements (lines 90-94).

- we have removed in our manuscript the phrase “data not shown” for fruit firmness at harvest (line 324) and the data of this parameter have been presented in Table 4.

- we have checked the reference list and cited papers in the text. We have added also 3 literature items added (17, 18, 22). For that reason, we had to change the numbering of papers in both the reference list and text. 

Reviewer #1: 

1) As suggested by the Reviewer, we should add information about time of the day when the tested spray treatments were applied. However, we would like to state that the above information was given in “The tested spray treatments and trial setup” (line 179) 

2) As suggested by the Reviewer, we should add an overview picture showing the tested treatments and results obtained. It seems that this suggestion is not justified because, in our opinion, the set-up of the experiment used is not complicated and the results presented in the tables and their description in the Results and Discussion sections are clear. 

 3) We agree with the Reviewer's argument to remove the phrase “with those nutrients” from Abstract (line 25).

Reviewer #2: 

1) As suggested by the Reviewer, we have added in the Introduction section information about studies related to biofortification of tree fruits with iodine and selenium (lines 64-69). 

2) Accordingly to the Reviewer’s suggestion, we have added information about pesticides and herbicides used and their application time (lines 130-138, 140-144). 

3) As proposed by the Reviewer, we have added information that the control trees were not sprayed with water (lines 171-172).

4) As suggested by the Reviewer, in the Result section, we should explain the results of all treatments whereas in the Discussion section, we should highlight the reasons of their result findings in the light of available literature. However, the Reviewer did not indicate which parts in the sections of Results and Discussion require the improvement. According to our opinion, in the Results section, we precisely described the impacts of the tested spray treatments on individual characteristics/parameters of plants. Moreover, in the Discussion section, we confronted the results obtained with those of the available scientific literature.

---

## [Editor Report · Decision Letter 1]

27 Feb 2023

Effects of preharvest sprays of iodine, selenium and calcium on apple biofortification and their quality and storability

PONE-D-22-35045R1

Dear Dr. Pawel Wojcik,

We’re pleased to inform you that your manuscript has been judged scientifically suitable for publication and will be formally accepted for publication once it meets all outstanding technical requirements.

Kind regards,

Hasan Sardar, Ph.D.

Academic Editor

PLOS ONE
---

## [Editor Report · Acceptance letter]

1 Mar 2023

PONE-D-22-35045R1 

Effects of preharvest sprays of iodine, selenium and calcium on apple biofortification and their quality and storability 

Dear Dr. Wójcik:

I'm pleased to inform you that your manuscript has been deemed suitable for publication in PLOS ONE. Congratulations! Your manuscript is now with our production department. 

Kind regards, 

on behalf of

Dr. Hasan Sardar 

Academic Editor

PLOS ONE